# Changes in temperature perception in transgender persons undergoing gender-affirming hormone therapy

Pauline Zimmermann [1], Martin Kaar[1], Theresa Bokeloh[1], Lotta Moll[1], Franziska Labinski[1], Falk Eippert [2], Matthias Blüher[1,3,4], Michael Stumvoll [1,3,4], Sascha Heinitz [1,3,4,5] & Haiko Schlögl [1,3,4,5] ✉

## Abstract

**Background** There are known sex disparities in temperature perception with lower thermal detection thresholds found in people assigned female at birth compared to people assigned male at birth. However, underlying mechanisms of these differences and the influences of sex hormones are not yet sufficiently understood.

**Methods** To assess the effects of sex hormones on temperature perception, we measured in a prospective observational cohort study temperature detection and pain thresholds with quantitative sensory testing and subjective temperature sensation in transgender patients undergoing gender-affirming hormone therapy (GAHT). We included 12 trans women (male-to-female transgender) and 17 trans men (female-to-male transgender) before and 3 and 6 months after start of GAHT. As a control group, we measured 13 cis women and 10 cis men without hormone treatment at the same timepoints.

**Results** Here we show that temperature detection thresholds in persons assigned female at birth at baseline are lower than in persons assigned male at birth. Accordingly, in trans women, temperature detection thresholds decrease with GAHT. Pain detection thresholds do not differ between sexes assigned at birth and do not change with time.

**Conclusions** We demonstrate that in trans women undergoing GAHT with estradiol and cyproterone acetate sensitivity to temperature changes increases, consistent with the greater temperature sensitivity observed in cis women compared to cis men. Future studies need to assess at which neurobiological processing stages the relevant changes occur and what molecular mechanisms play a role.

**Trial registration** NCT04838249.

## Plain english summary

Previous investigations showed that cis women can detect smaller temperature differences on their skin than cis men. However, it is not yet known if hormones play a role in these variations, or if, e.g., only different genes are responsible. To learn more about the influences of hormones on temperature perception, we tested how persons with different sex hormone levels can sense temperature. We measured 12 trans women (male-to-female transgender) and 17 trans men (female-to-male transgender) before and after 3 and 6 months of their hormone therapy. In trans women, temperature perception improved. These results help to better understand sex differences and hormone levels in temperature perception.

Sex disparities in temperature perception are known between people assigned female at birth compared to people assigned male at birth[1–8]. However, underlying mechanisms of these differences and the influences of sex hormones are not yet sufficiently understood[1]. Knowledge on temperature perception is important for multiple reasons: correct temperature sensing is needed to keep body temperature, a co-regulator of all physiological processes, in different environments in the optimal range, e.g., through vasomotor activity, sweating, shivering, or altered behavior[1,9].

Furthermore, thermal stimuli to the skin play an important role in thermal comfort[10]. Therefore, a further investigation of sex differences and the role of sex hormones is of particular interest[1].

Additionally to lower thermal detection thresholds, women show higher cold-wetness perception at the skin, are more likely to express thermal discomfort[11–15], feel cold and start shivering at higher temperatures than men[16] and have higher unpleasantness and pain intensity ratings for extreme temperatures[3,17]. Study results on the effects of sex hormones on

[1]Department of Medicine, University Hospital Leipzig, Leipzig, Germany. [2]Max Planck Insitute for Human Cognitive and Brain Sciences, Leipzig, Germany. [3]LeiCeM - Leipzig Center of Metabolism, Leipzig University, Leipzig, Germany. [4]Helmholtz Institute for Metabolic, Obesity and Vascular Research (HI-MAG) of the Helmholtz Zentrum München at the University of Leipzig and University Hospital Leipzig, Leipzig, Germany. [5]These authors contributed equally: Sascha Heinitz, Haiko Schlögl. ✉e-mail: haiko.schloegl@medizin.uni-leipzig.de

**Table 1 | Preparations and concentrations used in the study for gender-affirming hormone therapy**

| | 3 months | 6 months |
|---|---|---|
| **Trans women** | **n = 12** | **n = 11** |
| Estradiol | | |
| Estradiol gel 0.6 mg/g, 1.25−5 g/d | 9 | 8 |
| Estramon patch 75 µg/24 h | 3 | 3 |
| Cyproterone acetate 10 mg−12.5 mg/d | 12 | 11 |
| **Trans men** | **n = 17** | **n = 13** |
| Testosterone | | |
| Testosterone undecanoate injection 1000 mg, approx. every 3 months | 7 | 5 |
| Testosterone gel 16.2 mg/g, 1.25–2.5 g/d | 10 | 8 |

Hormone therapy was performed following international guidelines[27,28]. In trans women, estradiol was given transdermally via gel or patch; in trans men, testosterone was applied transdermally via gel or intramuscularly via injection following patient's preferences. Cyproterone acetate was given as tablet.

temperature signaling pathways are contradictory. Animal data suggest that the activity of cold receptor transient receptor potential melastatin (TRPM) 8 is directly modulated by testosterone[18], and castration of male mice increased sensitivity to mild cold[19]. While the effects of testosterone have been investigated more in depth, animal data on the molecular mechanisms mediating the action of estradiol on TRPM8 are scarce, and it is not yet clear if estradiol affects receptor function[20]. Few studies investigated the influence of sex hormones on TRPM2 receptors, which are involved in warm perception, in elaborate animal and in vitro models[21,22], but which have limited significance for physiology in healthy humans.

To our knowledge, no human study so far investigated direct effects of testosterone and estrogen on thermosensation[1]. Using a cohort undergoing gender-affirming hormone therapy (GAHT) offers the unique opportunity to prospectively assess the direct effects of testosterone and estradiol/cyproterone acetate during therapy intraindividually. This approach helps not only to disentangle the role of sex hormones in temperature perception and sensation in human physiology, but also to gain a better understanding of the effects of GAHT, which is highly relevant for the increasing number of trans persons seeking hormonal treatment[23,24].

We hypothesized that with quantitative sensory testing and a temperature sensation questionnaire we could detect changes in temperature perception and sensation in transgender persons undergoing GAHT[3]. We show for the first time to the best of our knowledge that during GAHT in trans women temperature detection thresholds decrease and suggest that sex hormone concentrations influence thermosensation.

## Methods
### Study cohort: transgender persons and control group
Patients with gender incongruence (diagnosis number after the international statistical classification of diseases and related health problems [ICD] −10 F64.0, ICD-11 HA60) were recruited at the endocrinological outpatient clinic of the University Hospital Leipzig. Measurements were performed between 01/2021 and 10/2023, as part of a larger trial monitoring various effects of GAHT (clinicaltrials.gov registration number NCT04838249). Twelve trans women (male-to-female transgender) and 17 trans men (female-to-male transgender) participated in the study. The study is observational, the decision for GAHT was solely based on patient's demand and clinical need, and independent from the study. The control group of persons without gender incongruence was recruited through both analog and online advertisements and measured between 01/2021 and 10/2023, comprising 13 cis women and ten cis men. Inclusion criteria for patients and control persons were age ≥18 years and for patients a signed informed consent about the GAHT. Exclusion criteria for both groups included serious medical conditions (e.g., uncontrolled high blood pressure, heart insufficiency, history of stroke, malignant diseases, or chronic infections), uncontrolled endocrine diseases (e.g., hypercortisolism, pituitary disease, hypo- or hyperthyroidism) and previous GAHT. All study participants consented to participating in the study.

Throughout the manuscript, we used the expression "sex" when referring to biological characteristics generally related to reproductive anatomy or physiology, while being aware that sex variables often include effects of gender[25]. We used "gender" when referring to culturally contextualized social experiences and expressions of identity[26]. For cisgender persons, i.e., persons who identify themselves with the gender that was assigned to them at birth, we are using the terms "cis women" and "cis men". When speaking of "persons assigned female at birth" we are referring to the pooled data from trans men before GAHT and cis women. Accordingly, "persons assigned male at birth" include pooled data from trans women before GAHT and cis men.

### Gender-affirming hormone therapy
Treatment was performed following international guidelines[27,28], with trans women receiving estradiol and cyproterone acetate ("testosterone blocker") and trans men receiving testosterone. Medication was controlled and adjusted following regular checks of blood concentrations of testosterone and estradiol. The preparations and concentrations that were used are listed in Table 1.

### Experimental design
Baseline measurements in patients were obtained immediately before the start and after 3 and 6 months of ongoing GAHT. In the control group, measurements were conducted at the same time intervals. The study protocol was identical for both patients and controls: in the morning, anthropometric data was assessed and a fasting blood draw was performed. Body surface was calculated by Mosteller formula[29]. Body composition (body fat mass, body fat free mass) was determined using body impedance analysis[30]. At approximately 11 a.m., participants had lunch and at approximately 12:30 pm quantitative sensory testing was performed in the outpatient clinic of the Clinic for Neurology at the University Hospital Leipzig. Room and skin temperature at the hand were measured after entering the room. If the skin temperature was below 28 °C, participants were instructed to hold their hands under warm water for 20 s, after which the temperature was measured again. After that, participants had to complete a questionnaire addressing temperature sensation (see Temperature sensation questionnaire, Supplementary Fig. S1). Then, quantitative sensory testing was performed as described below, and skin temperature of the hand was measured again after removing the thermode.

### Quantitative sensory testing
Quantitative sensory testing was performed using the Main Station Neurosensory Analyzer Model TSA-II (2001), Medoc, Ramat Yishay, Israel with its standard 30×30 mm thermode (Peltier-element) with temperature sensors and an accuracy of 0.3 °C[31]. All study staff underwent training and followed a standardized study protocol for both delivering verbal instructions to volunteers as well as performing the quantitative sensory testing procedures. Procedures always took place at the same test location and during the same daytime and followed the same order. The thermode was placed at the palm as suggested by previous literature[32–35] of the non-dominant hand and fixed with velcro tape. Participants were instructed to press a button with their dominant hand as soon as they detected a change in temperature at the site of thermal stimulation. In the first set of measurements, the thermode's temperature decreased according to the method of limits with a rate of 0.3 °C/s from its baseline temperature set at 32 °C until the participant pressed the button to measure cold detection thresholds. In the second set of measurements, the thermode's temperature increased accordingly to measure warm detection thresholds. Four measurements for

**Table 2 | Baseline characteristics and laboratory data of the study population**

| | Assigned male at birth | | | Assigned female at birth | | |
|---|---|---|---|---|---|---|
| Characteristics | Trans women (*n* = 12) | Cis men (*n* = 10) | *p* | Trans men (*n* = 17) | Cis women (*n* = 13) | *p* |
| Age (years) | 25.2 (6.1) | 24.8 (3.9) | 0.54 | 22.6 (4.7) | 24.8 (4.1) | **0.05** |
| Weight (kg) | 74.4 (16.5) | 76.9 (10.4) | 0.63 | 71.5 (20.7) | 65.7 (16.4) | 0.12 |
| Body mass index (kg/m²) | 23.5 (6.1) | 23.7 (4.0) | 0.92 | 25.9 (7.0) | 23.0 (2.9) | 0.17 |
| Body surface (m²) | 1.92 (0.21) | 1.98 (0.19) | 0.38 | 1.83 (0.23) | 1.76 (0.29) | 0.16 |
| Body fat mass (kg) | 16.1 (14.1) | 14.7 (4.4) | 0.97 | 25.5 (13.3) | 17.4 (7.0) | **0.04** |
| Body fat free mass (kg) | 59.1 (4.3) | 62.75 (8.7) | 0.35 | 47.7 (7.3) | 47.8 (9.1) | 0.64 |
| Testosterone concentration (nmol/l) | 15.40 (5.82) | 18.47 (8.0) | 0.20 | 1.06 (0.5) | 0.98 (0.7) | 0.36 |
| Estradiol concentration (pmol/l) | 90.3 (36.2) | 105.5 (75.9) | 0.34 | 222.0 (234.0) | ª387.0 (269.5) | 0.18 |

Median (interquartile range) and two-sided Mann-Whitney-U test for not-normally distributed data. *P*-values ≤ 0.05 are displayed in bold.
ª*n* = 9; 4 cis women, who took hormonal contraceptive with ethinylestradiol were excluded for this specific calculation.

cold perception and four for warm perception were conducted and the respective means were calculated and used for further analyses. Before the next sets of measurements, participants were instructed to press the button as soon as they felt a heat or cold induced pain. Again, starting from the skin indifference temperature of 32 °C, the thermode's temperature decreased/increased once at a rate of 1.5 °C/s until the button was pressed or the lowest possible temperature of 0 °C/highest possible temperature of 50 °C was reached[31]. We did not conduct further measurements of mechanical (pain) thresholds or measurements of different test sites.

**Temperature sensation questionnaire**
Participants answered an 18-question-long questionnaire based on "The experienced temperature sensitivity and regulation survey"[36]. Statements regarding temperature sensation were rated by participants on a numeric rating scale ranging from zero to ten. The questions were designed to assess temperature sensations in everyday situations and to quantify the experience of hot and cold flushes. All questions and possible answers, as well as how questions were grouped for calculation of the two summary scores are listed in the Supplemental methods, Supplementary Fig. S1.

**Statistics**
An a priori power analysis was not conducted, as the expected effect sizes could not be estimated and the number of patients willing to participate could not be determined in advance. Instead, all patients who consented to participate during the study period and met the inclusion criteria were enrolled. Four cis women who were treated with hormonal contraceptives with ethinylestradiol during the measurements were excluded from the analysis of estradiol concentrations, as ethinylestradiol suppresses endogenous estradiol secretion and is not detected by the estradiol laboratory assay used, leaving nine participants in this subgroup. Due to small group sizes (<20 persons) we used non-parametric tests throughout the measurements (Mann-Whitney-U test, mixed model analyses with GraphPad Prism, Dotmatics, Boston/USA, version 10.0.2, Durbin-Skillings-Mack[37] test and Conover pairwise rank comparison[38] with XLSTAT, Lumivero, Denver/USA, version 2023.3.1-1416). Additionally, we performed Friedman's and Dunn's test under exclusion of participants with missing data and results are reported in the Supplemental material. Global tests for the two a priori defined primary outcomes - temperature detection thresholds cold and warm - were conducted without multiple-comparison correction. For *post-hoc* pairwise comparisons, p-values were adjusted using the Bonferroni correction by dividing the nominal alpha level (0.05) by the number of pairwise tests (3), resulting in an adjusted significance threshold of p < 0.0167. To test for equivalence, we used equivalence paired t-tests using JASP, University of Amsterdam, Amsterdam/The Netherlands, version 0.95 (bound specifications in raw, equivalence region from −0.05 to 0.05, alpha level 0.05). Correlation analyses between temperature detection thresholds and summary scores cold and warm, age, body mass index (BMI), fat mass, fat free mass, room temperature and average monthly temperature were

performed with GraphPad Prism using Spearman's correlation including all baseline measurements. Correlation analyses of fat mass and fat free mass were separately calculated for assigned female and male sex at birth due to sex differences in body composition. Correlation analyses between changes in temperature detection thresholds and changes in serum testosterone and estradiol concentrations over the study period of six months were performed with GraphPad Prism using Spearman's correlation.

## Results
### Anthropometric and laboratory parameters
Baseline characteristics are summarized in Table 2. As expected, in trans women testosterone serum concentrations decreased during 6 months of GAHT ($p \le 0.0001$, Kendall's $W = 0.80$; Conover's procedure: 0 vs. 3 months $p \le 0.05$, 0 vs. 6 months $p \le 0.05$, 3 vs. 6 months $p \ge 0.05$) and estradiol concentrations increased ($p = 0.001$, Kendall's $W = 0.59$; Conover's procedure: 0 vs. 3 months $p \le 0.05$, 0 vs. 6 months $p \le 0.05$, 3 vs. 6 months $p \ge 0.05$). In trans men, testosterone concentrations increased ($p \le 0.0001$, Kendall's $W = 0.75$; Conover's procedure: 0 vs. 3 months $p \le 0.05$, 0 vs. 6 months $p \le 0.05$, 3 vs. 6 months $p \ge 0.05$), while we found no changes in serum estradiol concentrations ($p = 0.60$, Kendall's $W = 0.02$). In cis men and cis women, neither in testosterone ($p = 0.47$, Kendall's $W = 0.11$ and $p = 0.74$, Kendall's $W = 0.04$), nor estradiol concentrations ($p = 0.76$, Kendall's $W = 0.05$ and $p = 0.73$, Kendall's $W = 0.08$) we found significant changes over time. In the interaction analysis between trans women and cis men, significant differences were observed in testosterone ($p \le 0.0001$) and estradiol ($p = 0.03$) concentration changes over the study period. Between trans men and cis women, we observed significant differences in testosterone ($p \le 0.0001$), but not in estradiol concentration changes ($p = 0.58$) (Fig. 1).

### Quantitative sensory testing for temperature perception
Correlation analyses including all baseline measurements showed no significant correlations between cold and warm detection thresholds and age, BMI, fat mass, fat-free mass and the average monthly temperatures at the time of the measurements (all $p > 0.05$). Room temperature correlated with warm ($r = -0.32$, $p = 0.03$) (Supplementary Fig. S2A), but not with cold detection thresholds ($r = 0.17$, $p = 0.26$).

In quantitative sensory testing, median (interquartile range) threshold values for cold and warm detection at baseline measurements were in persons assigned female at birth (trans men before GAHT treatment and cis women at baseline) lower ($-1.04$ [0.73] and 1.31 [0.74] °C) than in persons assigned male at birth (trans women before GAHT treatment and cis men at baseline) ($-1.50$ [1.34] and 1.83 [0.81] °C) ($p = 0.006$ and $p = 0.01$; Cohen's $d = 0.73$ and 0.74) (Fig. 2A).

In trans women, over the time period of 6 months of GAHT, the thresholds for both cold ($p = 0.004$, Kendall's $W = 0.63$) and warm ($p = 0.045$, Kendall's $W = 0.31$) detection decreased significantly. Measurements for the specific time points (0 vs. 3, 0 vs. 6, and 3 vs. 6 months)

**Fig. 1 | Serum testosterone and estradiol concentrations.** Testosterone and estradiol concentrations in trans women, cis men, trans men, and cis women grouped by assigned sex at birth. Median ± interquartile range. Statistical testing: Durbin-Skillings-Mack test to analyze changes over time; mixed model analysis, two-sided, for analysis between subgroups. *P*-values for group comparisons between MtF and cis men; FtM and cis women of testosterone $p < 0.0001$; $p < 0.0001$ and estradiol $p = 0.029$; $p = 0.581$. *P*-values of changes over time of testosterone, estradiol for MtF $p < 0.0001$, $p = 0.001$; cis men $p = 0.471$, $p = 0.762$; FtM $p < 0.0001$, $p = 0.602$; and cis women $p = 0.743$, $p = 0.725$. Sample sizes reported in the Figure are at Baseline assessment (0 months). Sample sizes for 3, 6 months were for MtF; cis men; FtM; and cis women, respectively, for testosterone 12, 11; 9, 9; 17, 13; and 12, 10, and for estradiol 12, 11; 9, 9; 17, 13; and 8, 7. Four cis women with hormonal contraceptives were excluded from the analysis of estradiol concentrations. FtM female-to-male transgender, MtF male-to-female transgender, *$p \le 0.05$, ***$p \le 0.001$, ****$p \le 0.0001$.

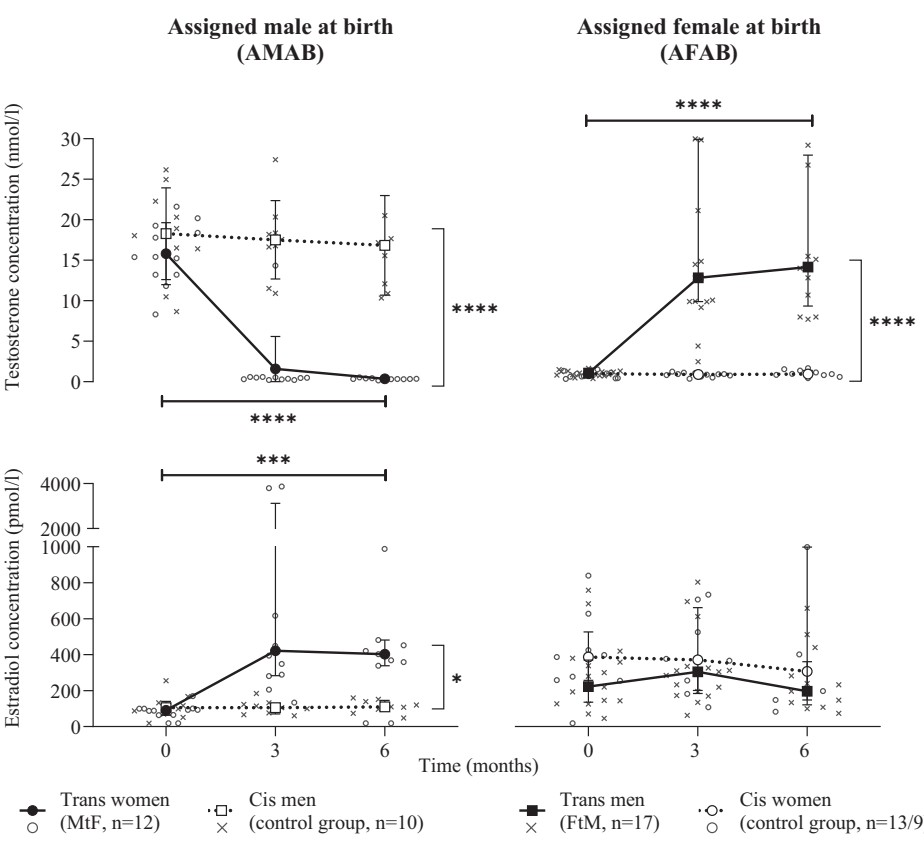

were not significantly different in the *post hoc* analysis (all $p > 0.05$). We found no changes in cold and warm temperature detection thresholds in trans men ($p = 0.75$, Kendall's $W = 0.06$ and $p = 0.82$, Kendall's $W = 0.02$), cis women ($p = 0.95$, Kendall's $W = 0.02$ and $p = 0.19$, Kendall's $W = 0.13$), and cis men ($p = 0.10$, Kendall's $W = 0.20$ and $p = 0.29$, Kendall's $W = 0.30$) (Fig. 2B). In the interaction analysis, neither between trans women and cis men there were significant differences in cold and warm threshold changes ($p = 0.89$ and $p = 0.98$), nor between trans men and cis women ($p = 0.54$ and $p = 0.58$). Equivalence was not demonstrated in any group (Supplementary Table S1). Correlation analyses between changes in cold and warm detection thresholds and serum testosterone and estradiol concentrations did not reveal significant associations (Supplementary Fig. S3).

Mann-Whitney-U and Wilcoxon test, respectively, did not show statistically significant differences in room temperature at measurements between sexes assigned at birth at baseline and in trans women over the three measurements in 6 months (all $p > 0.05$).

We did not find differences for temperature thresholds for cold pain sensation between the sexes assigned at birth ($p = 0.51$, Cohen's $d = 0.33$), nor did we observe changes among trans women, trans men, cis women, or cis men across the study period ($p = 0.28$, Kendall's $W = 0.16$; $p = 0.37$, Kendall's $W = 0.06$; $p = 0.24$, Kendall's $W = 0.11$; and $p = 0.56$, Kendall's $W = 0.04$). For heat pain, we found no differences between the sexes assigned at birth ($p = 0.43$, Cohen's $d = 0.11$), nor did we observe changes among trans women, trans men, cis women, or cis men across the study period ($p = 0.91$, Kendall's $W = 0.000$; $p = 0.26$, Kendall's $W = 0.08$; $p = 0.90$, Kendall's $W = 0.01$; and $p = 0.08$, Kendall's $W = 0.44$) (Supplementary Fig. S4). Equivalence was not demonstrated in any group (Supplementary Table S1).

**Temperature sensation questionnaire**
Correlation analyses showed no correlations between summary score cold and age, BMI, fat mass, fat free mass, current room temperature and the average monthly temperatures at the time of the measurements (all

$p > 0.05$). Summary score warm showed no correlation with age, fat mass, fat free mass, current room temperature and the average monthly temperatures at the time of the measurement (all $p > 0.05$), but a positive correlation with BMI ($r = 0.32$, $p = 0.02$) (Supplementary Fig. S2B).

At baseline, we did not find differences in summary scores for cold and warm feelings in daily situations between persons assigned female and male at birth ($p = 0.22$, Cohen's $d = 0.22$; $p = 0.46$, Cohen's $d = 0.05$). Moreover, we observed no changes over 6 months of GAHT in summary scores for cold (trans women: $p = 0.17$, Kendall's $W = 0.20$; trans men: $p = 0.27$, Kendall's $W = 0.16$; cis women: $p = 0.34$, Kendall's $W = 0.12$; cis men: $p = 0.82$, Kendall's $W = 0.05$) or warm (trans women: $p = 0.36$, Kendall's $W = 0.14$; trans men: $p = 0.78$, Kendall's $W = 0.02$; cis women: $p = 0.20$, Kendall's $W = 0.17$; cis men: $p = 0.59$, Kendall's $W = 0.11$) (Supplementary Fig. S5). Equivalence was not demonstrated in any group (Supplementary Table S1).

To estimate if there was a link between temperature detection thresholds and summary scores cold and warm, we correlated values of cold and warm detection thresholds with values of summary scores cold and warm, including all baseline measurements. Neither did we find a significant correlation between cold detection thresholds and summary scores cold ($p = 0.42$), nor between warm detection thresholds and summary scores warm ($p = 0.15$) (Supplementary Fig. S2C, D).

Results of the evaluation of the room temperature and the occurrence of cold and hot flushes are presented in the Supplemental material in the Supplemental results section (p. 6, ll. 87–100). Also, all statistical analyses calculated with Friedman's and Dunn's test under exclusion of participants with missing data can be found in the Supplemental material in the Supplemental results section (p. 6, l. 102 – p. 8, l. 136). These control analyses qualitatively supported the results from the main analyses.

## Discussion
When measuring temperature detection thresholds using quantitative sensory testing, we saw lower cold and warm detection thresholds in persons

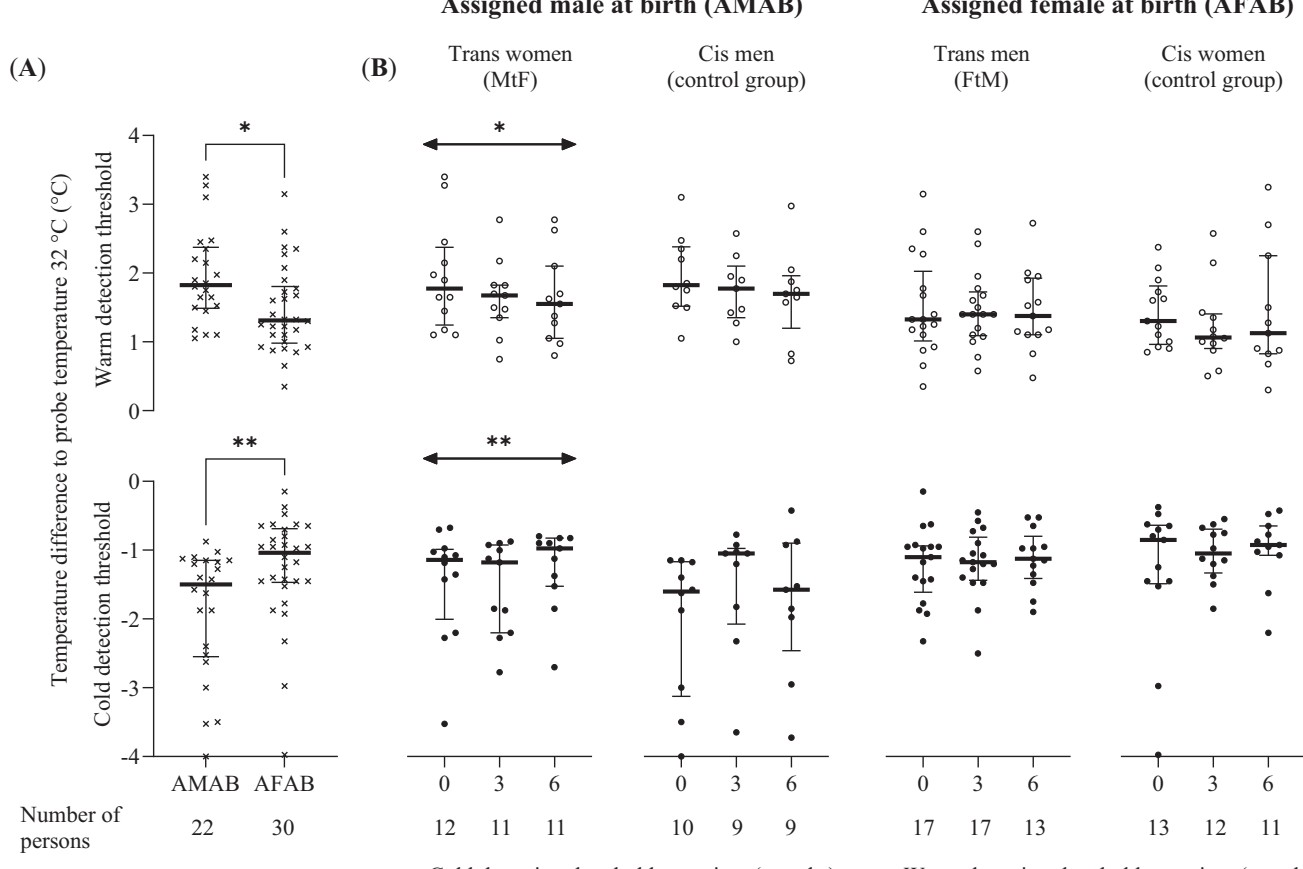

**Fig. 2 | Temperature detection thresholds measured by quantitative sensory testing.** Median ± interquartile range of temperature detection thresholds (**A**) grouped by assigned sex at birth and (**B**) in trans women, cis men, trans men, and cis women. Statistical testing: (**A**) Mann-Whitney-U test, two-sided, (**B**) Durbin-Skillings-Mack test. *P*-values for warm, cold detection thresholds: (**A**) *$p$ = 0.010, **$p$ = 0.005; (**B**) MtF $p$ = 0.045, $p$ = 0.004; cis-men $p$ = 0.283, $p$ = 0.097; FtM $p$ = 0.818, $p$ = 0.747; cis-women $p$ = 0.189, $p$ = 0.946. Assigned female at birth: all data from trans men before GAHT and cis women at baseline, assigned male at birth: all data from trans women before GAHT and cis men at baseline. FtM female-to-male transgender, GAHT gender-affirming hormone therapy, MtF male-to-female transgender, *$p ≤ 0.05$, **$p ≤ 0.01$.

assigned female at birth compared to male at birth, which is in accordance with previous literature[2–7]. Furthermore, we saw a threshold reduction over the observation period of six months in trans women treated with estradiol and cyproterone acetate.

A possible mechanism leading to the observed effects is that testosterone interacts with the androgen receptor and with the cold receptor TRPM8 and reduces the channel activity of TRPM8. This is supported by a study in mice showing that castration-induced hypogonadism in males led to heightened sensitivity to mild cold[19]. Similar to this effect, the testosterone blocker cyproterone acetate could lead to an increased sensitivity in temperature changes. However, data on the effects of testosterone on TRPM8 is inconclusive; e.g., in a cell culture study, testosterone activated the cold receptor TRPM8 and opened it completely in lipid layers[18].

Estrogen may also modulate receptor signaling[39]. E.g., one study found that during the luteal phase of the menstrual cycle—when progesterone and estrogen concentrations are higher than in the follicular phase—the threshold for cold perception is lower[40]. This finding was confirmed by another study with a larger cohort, although only at the mammilla[41]. However, data on the effects of estradiol on cold receptors is scarce and it is not yet clear how the molecule affects receptor function[20].

The role of estrogen on central nervous thermoregulation was investigated in several studies. Data suggests that estrogen affects thermoregulatory pathways in different brain regions. These include the medial preoptic area and the ventromedial nucleus of the hypothalamus, with heterogenous effects on body temperature that may depend on the neuron type, the species or other modulatory inputs[42–44]. These effects of estrogen are used in postmenopausal treatment of vasomotor symptoms as, e.g., hot flushes and night sweating[45,46].

Another mechanism that could contribute to lower temperature detection thresholds in trans women undergoing GAHT may be the skin softening effect of therapy. In cis persons, skin thickness is different in men compared to women[47]. After start of GAHT, softening of skin is observable already after three to six months[27,48]. In a study using a thermal model describing the temperature evolution in skin, thicker skin seemed to decrease thermal perception[49]. However, another study investigating only male participants did not find associations between skin thickness at the fingertip and thermotactile perception[50,51]. Bodily changes in trans women under GAHT, e.g., breast growth, redistribution of body fat or decrease of the growth rate of body hair have a visible onset after 3 to 6 months and reach their maximum after 2–5 years of GAHT[27,48]. Having investigated the first 6 months of GAHT, we cannot say whether the changes in temperature perception in trans women will persist during lifelong treatment, increase, or vanish after adaption to the new hormonal situation.

In trans men who were treated with testosterone, we did not see significant changes in temperature perception. In this group, testosterone serum concentrations increased from the female reference range to within the male reference range through therapy. However, usually during the first 6 months of treatment with testosterone, the hypothalamic-pituitary-gonadal axis is not yet suppressed and still gonadotropin-releasing hormone, luteinizing/follicle-stimulating hormone and consequently estrogens

are secreted. The process of hypothalamic-pituitary-gonadal axis suppression though testosterone therapy with inhibition of ovulation and menstrual bleeding, and accompanying decrease of ovarian estrogen secretion in many patients takes longer[27]. Also, in our sample of trans men, median estradiol concentrations had not significantly declined after 6 months of treatment compared to baseline.

When measuring temperature sensation using the summary scores cold and warm, we saw neither differences in summary scores cold and warm between persons assigned female and male at birth, nor changes during the study period of 6 months in the four groups, nor correlations between the results of quantitative sensory testing and the temperature sensation questionnaires. As stated in our methods, we only used a subset of the original 21 dimensions of the questionnaire, which were sensitive to sex-differences. Therefore, the questions we used might not be suitable for measuring sex-differences, or our cohort might have been too small.

We found no differences in temperature thresholds for pain sensation ("cold pain" or "heat pain") between the sexes assigned at birth and no changes in any of the four groups over the study period. Previous literature regarding sex differences in pain sensation is heterogeneous, one study found sex differences[3], while another found sex differences in heat pain only[52], and one found no differences at all[53]. Here, more research will be necessary, including inter-individual differences and other co-factors rather than sex[25,26].

For our study, we chose a design with four groups, consisting of trans women and trans men undergoing GAHT, and cis women and cis men without hormone treatment, and performed statistical tests for group differences in order to investigate the effects of GAHT and compare with untreated persons. However, we are aware that there is a lot of important work published on the necessity of elucidating biological mechanisms outside of binary sex categories[25,26]. The prior assumption of binarity may not be true for all biological mechanisms of the human body and thus lead to biased results. Further investigations without this prior assumption should be made in larger samples with the statistical power for respective analyses.

Limitations to our study include that, first, our group sizes are relatively small for conducting a clinical study. However, the median temperature detection thresholds in our group were 0.3 °C lower in persons assigned female at birth than in persons assigned male at birth, which was in the range of 0.2–0.4 °C described before in the cohort of 1252 participants[3]. Second, as our results did not withstand the interaction analysis, we cannot be sure that GAHT was causal for observed effects, but measurements in trans persons may have changed over the study period for different reasons (e.g., habituation to the study procedures). Nevertheless, this study tried to minimize this limitation as control groups underwent the same measurements in the same time intervals.

Taken together, our findings are especially of importance, since there is an increase in the prescription of GAHT[23,24] in transgender persons and our findings help to gain further knowledge of changes in body physiology in persons undergoing hormone therapy. Considering that thermosensation also influences body temperature regulation[1], a better understanding of how sex hormones influence temperature sensation is also important for cisgender persons exposed to more extreme temperatures due to climate change. To investigate underlying mechanisms which are causal for the observed changes in temperature perception, subsequent studies are necessary.

## Conclusions
Taken together, in our study, we show that temperature detection thresholds decreased in persons with initially male testosterone serum concentrations during treatment with estradiol and cyproterone acetate, and thus transitioned to a female hormonal situation. With our study setup and methods, in trans women under GAHT, we found changes over time in temperature perception that were comparable to the differences between sexes assigned at birth. Future studies are needed to assess the mechanisms behind these changes to gain a better physiological understanding of how sex hormones affect temperature perception.

## Ethics approval and consent to participate
The work described in this article has been carried out in accordance with The Code of Ethics of the World Medical Association (Declaration of Helsinki). The Ethics Committee of the University of Leipzig approved this research project (approval no. 023/20-ek), according to the national research ethics regulations. All participants gave their written consent for all study procedures.

## Data availability
The source data for Figs. 1 and 2 is in Supplementary Dataset 1. Other datasets generated and/or analyzed during the current study are not publicly available due to patient confidentiality. The corresponding author will on request detail the restrictions and any conditions under which access to some data may be provided.

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

## Acknowledgements

We thank all participants for their participation in the study. We thank Jens Przybilla and Paul Czechowski for statistical advice. We thank Natalia Schischkarjow, Björn Drechsler-Kryst, Lotte Oldenburg and Antonia Stengler for helping to conduct the study. Pauline Zimmermann and Franziska Labinski received a 6-month scholarship from the German Diabetes Society (*Deutsche Diabetes Gesellschaft e.V.*) during their medical studies. The study was supported by the Deutsche Forschungsgemeinschaft (DFG, German Research Foundation) under Germany´s Excellence Strategy—EXC-3105/1— 533765739 (to M.S.) and a research grant from Besins Healthcare (to H.S.). The funders had no role in the study design, data collection, data analysis, interpretation of results, or

the writing of the manuscript. Coverage of the publication fee was supported by the Open Access Publishing Fund of Leipzig University.

## Author contributions

P.Z.: data analysis, conduction of the study, writing, reviewing and editing of the manuscript; M.K.: conduction of the study, reviewing and editing of the manuscript; T.B.: conduction of the study, reviewing and editing of the manuscript; L.M.: conduction of the study, reviewing and editing of the manuscript; F.L.: conduction of the study, reviewing and editing of the manuscript; F.E.: reviewing and editing of the manuscript; M.B.: funding acquisition, resources, reviewing and editing of the manuscript; M.S.: funding acquisition, resources, reviewing and editing of the manuscript; S.H.: conceptualization, conduction of the study, reviewing and editing of the manuscript; H.S.: conceptualization, funding acquisition, data analysis, conduction of the study, project administration, writing, reviewing and editing of the manuscript. All authors read and approved the final manuscript.

## Funding

## Competing interests

H.S. received financial support for this study from Besins Healthcare. The funder had no role in the study design, data collection, data analysis, interpretation of results, or the writing of the manuscript. M.B. received honoraria as a consultant and speaker from Amgen, AstraZeneca, Bayer, Boehringer-Ingelheim, Lilly, Novo Nordisk, Novartis, and Sanofi. All other authors have no competing interests.
