## [Transparent Peer Review file · Communications Medicine]

Changes in temperature perception in transgender persons undergoing gender-affirming hormone therapy

Corresponding Author: Dr Haiko Schlögl

Version 0:

Reviewer comments:

Reviewer #1

(Remarks to the Author)

Reviewer #2

(Remarks to the Author)

In this research, the authors examined the effect of gender-affirming hormone therapy (GAHT) on temperature perception in transgender persons, as well as cis women/men (as control). Improvement in temperature detection ability was only found in trans women during GAHT, not in other groups (trans men, cis women/men). The methodology of this study is generally appropriate and the findings are interesting. However, several aspects of the analysis and discussion warrant further clarification and improvement.

Major issue:

1. The study measured the serum testosterone and estradiol concentrations under hormone therapy, and one's temperature detection thresholds. It would be informative to combine these two sets of results and examine whether one's changes in temperature perception are correlated with changes in serum testosterone/estradiol concentrations. This analysis could partially address the statement in highlights ("Sex hormones testosterone and estradiol might be relevant factors for sex differences in temperature perception and sensation.")
2. In addition to quantitative measurement of temperature sensation, the study also assessed subjective temperature sensation, which showed no difference between groups. Are there any correlations between the objective and subjective measurements of temperature sensation? Furthermore, the discussion section did not address the potential reasons for the dissociation between these two measurements.

Minor issue:

1. One of the highlights was "In persons assigned female at birth in the absence of hormone treatment, temperature detection thresholds were lower than in persons assigned male at birth". However, in the first paragraph of the discussion, the authors indicated that this result was in accordance with previous literature (citing six studies), which makes it seem more like a confirmation of existing findings results rather than a novel highlight.
2. In the discussion section and conclusion, the authors indicated that trans women's treatment were "estrogen and testosterone blockers". According to Table 2, trans women received Estradiol and Cyproteroneacetate (testosterone blocker) for gender-affirming hormone therapy. It would be more precise to use the singular form of "blocker" since there was only one kind of testosterone blockers.
3. In the Figure 2a, the authors should denote what the triangle and the square represent.

Reviewer #3

(Remarks to the Author)

This study investigates sex hormone influences on thermosensation in transgender individuals undergoing GAHT. The study design is longitudinal with three data points (0, 3 and 6 months) with an appropriate cisgender control group.

Thermosensation was assessed using a quantitative sensory testing protocol that included the assessment of detection and pain thresholds, as well as a questionnaire.

The main finding is that feminizing GAHT in trans women leads to reduced detection thresholds, comparable with those observed in cisgender women compared to cisgender men.

Main comments

- There is no sample size justification or prior power analysis reported.
- Effect sizes are not reported.
- There are several null effects, but it's not possible to have an insight on whether these are true null effects or the study is underpowered. Equivalence testing or Bayesian statistics would provide some insight on how to interpret the null effects.
- Correction for multiple comparisons is not mentioned.
- Specific instructions given to participants should be included (e.g., were the instructions the same as in the standardised QST manual or were they different). On this note, it seems that the protocol followed the German standardised QST protocol, but there is no explicit mention of whether it was the case or what are the specific differences that were introduced. For instance, were there familiarisation trials? The analyses says that 4 trials were analysed, but if the standardised version of the protocol was used, the first one is about familiarisation/demonstration, while the 2nd to 4th trials are the test trials to analyse

Minor comments

- "In trans women, over the period of 6 months of GAHT, the thresholds for both "cold" ($p=0.004$) and "warm" ($p=0.045$) detection decreased (both $p>0.05$ in post-hoc tests)." This statement needs clarification: does this mean that the overall change over the 6 month period was statistically significant, but the specific time points (0 vs 3 and 3 vs 6) were not?
- There are instances of phrases like "temperature detection improved..." that should be avoided because they imply causality, which cannot be concluded from an observational design (i.e., no randomisation or placebo group). This could be rephrased to "was associated with..."
- "discriminating "cold" and "warm" (page 9): the authors should be more careful with the use of the word "discrimination", as that is a specific term associated with 2 AFC/IFC tasks, where the participants need to discriminate between two stimuli. Instead, the task used by the author is about "detection" of a single stimulus.
- In figures 2, it's not obvious what the first and second rows of results represent. It took me quite a while to figure out that they are WDT (first row) and CDT (second row) based on the color of the dots
- It would be more complete to also see a figure about cold and heat pain thresholds. Also the non-significant results about pain thresholds should be reported as they could be useful for future meta-analyses

Version 1:

Reviewer comments:

Reviewer #2

(Remarks to the Author)

I have reviewed the authors' responses to the previous comments and the changes made to the manuscript. I find the revisions satisfactory and I have no further comments.

Reviewer #3

(Remarks to the Author)

The authors have addressed all my previous comments.

Answers to reviewers' comments

(page and line numbers refer to the manuscript version with tracked changes)

Reviewer #1:

Provide more detail to the manuscript, citing supporting references and clearly stating the aim of the study.

We thank the reviewer very much for reviewing our manuscript! We have now restructured the introduction of our manuscript and are citing more supporting references for sex differences in temperature perception (references¹⁻⁸) and are stating the aim of our study in more detail (throughout entire introduction, pp. 3-4, ll. 78-141).

The term “improved” is inaccurate and should be changed throughout, as this suggests AFAB people need improvement or are lacking in some way. More accurate descriptive language should be used, e.g. becomes more sensitive to heat, cold, pain, indicating levels of sensitivity to high or low temps, “feminising GAHT” instead of “estrogen”, etc.

Thank you for this valuable comment! We changed our wording accordingly and we are now speaking of “lowered” or “decreased temperature detection thresholds” and “sensitivity to temperature changes increases”, rather than an “improvement” of perception (e.g., p. 2, ll. 58-63). We also now use accurate descriptive language as you suggested and speak of “testosterone” and “estradiol and cyproterone acetate” (e.g., p. 2, ll. 58-63) when describing the treatment of our patients. When not speaking of specific estrogens (like estrone, estradiol, estriol, or estetrol) we use the term “estrogen” (e.g., p. 4, l. 125).

Rather than detailing the mechanisms at play when going from descriptions of AFAB to transgender women authors rely on the category of AFAAB to do the biological work - there is a lot of critical work published on the necessity of elucidating biological mechanisms - especially since that it the aim of the study (e.g. DuBois and Shattuck Heidorn 2021, Springer et al 2012). Authors should be careful not to confuse sex and gender.

Thank you very much for these helpful comments! We now included the following section in our discussion:

“For our study, we chose a design with four groups, consisting of trans women and trans men undergoing GAHT, and cis women and cis men without hormone treatment, and performed statistical tests for group differences in order to investigate effects of GAHT and compare with untreated persons. However, we are aware that there is a lot of important work published on the necessity of elucidating biological mechanisms outside of binary sex categories. The prior assumption of binarity may not be true for all biological mechanisms of the human body and thus lead to biased results. Further investigations without this prior assumption should be made in larger samples with the statistical power for respective analyses” (p. 14, ll. 466-473).

We now use the definitions of sex and gender as proposed by DuBois and Shattuck Heidorn 2021 as suggested by the reviewer:

“Throughout the manuscript, we used the expression “sex” when referring to biological characteristics generally related to reproductive anatomy or physiology, while being aware that sex variables often include effects of gender. We used “gender” when referring to culturally contextualized social experiences and expressions of identity” (p. 7, ll. 162-165).

Discussion of the work on menopause and the role of estrogen therapy on temperature regulation should be included, such as adaptive biology regarding temperature rather than recapitulating characterizations of biological processes.

We shortened the recapitulation of the biological mechanisms (pp. 3, 4, ll. 79-123) and inserted a passage in the discussion regarding the role of estrogen on thermoregulatory pathways in the brain as well as the effects of estrogen therapy on thermoregulation in menopause:

“The role of estrogen on central nervous thermoregulation was investigated in several studies. Data suggests that estrogen affects thermoregulatory pathways in different brain regions. These include the medial preoptic area and the ventromedial nucleus of the hypothalamus, with heterogenous effects on body temperature that may depend on the neuron type, the species or other modulatory inputs. These effects of estrogen are used in postmenopausal treatment of vasomotor symptoms as, e.g., hot flushes and night sweating” (p. 17, ll. 422-427).

Also, we inserted a review regarding physiological processes in thermoregulation:

“Knowledge on temperature perception is important for multiple reasons: correct temperature sensing is needed to keep body temperature, a co-regulator of all physiological processes, in different environments in the optimal range, e.g., through vasomotor activity, sweating, shivering or altered behaviour.” (p. 4, ll. 106-111).

Reviewer #2:

In this research, the authors examined the effect of gender-affirming hormone therapy (GAHT) on temperature perception in transgender persons, as well as cis women/men (as control). Improvement in temperature detection ability was only found in trans women during GAHT, not in other groups (trans men, cis women/men). The methodology of this study is generally appropriate and the findings are interesting. However, several aspects of the analysis and discussion warrant further clarification and improvement.

We thank the reviewer very much for reviewing our manuscript! And thank you very much for acknowledging that our methodology is appropriate and the findings interesting!

Major issue:

1. *The study measured the serum testosterone and estradiol concentrations under hormone therapy, and one's temperature detection thresholds. It would be informative to combine these two sets of results and examine whether one's changes in temperature perception are correlated with changes in serum testosterone/estradiol concentrations. This analysis could partially address the statement in highlights ("Sex hormones testosterone and estradiol might be relevant factors for sex differences in temperature perception and sensation.")*

Thank you for this very good idea! We performed the suggested analysis and correlated changes in testosterone and in estradiol concentrations with changes in warm and cold detection thresholds from baseline to the measurement at 6 months including all four study groups and inserted this paragraph under results:

"Correlation analyses between changes in cold and warm detection thresholds and serum testosterone and estradiol concentrations did not reveal significant associations". (p.13, ll. 311-313).

The results are presented in Suppl. Figure S3.

2. *In addition to quantitative measurement of temperature sensation, the study also assessed subjective temperature sensation, which showed no difference between groups. Are there any correlations between the objective and subjective measurements of temperature sensation? Furthermore, the discussion section did not address the potential reasons for the dissociation between these two measurements.*

Thank you also for this excellent idea! We performed the suggested analysis and correlated values of warm and cold detection thresholds with the summary scores warm and cold including all baseline measurements. We included the following paragraph in the results section:

"To estimate if there was a link between temperature detection thresholds and summary scores cold and warm, we correlated values of cold and warm detection thresholds with values of summary scores cold and warm including all baseline measurements. Neither did we find a significant correlation between cold detection thresholds and summary scores cold ($p=0.42$), nor between warm detection thresholds and summary scores warm ($p=0.15$) (Suppl. Figure S2C, D)" (p. 14, ll. 352-357)

and added Suppl. Figures S2C and D to the Supplement. We are now discussing the outcomes of this analysis together with potential reasons for the dissociation between these two measurements:

"When measuring temperature sensation using the summary scores cold and warm, we saw neither differences in summary scores cold and warm between persons assigned female and male at birth, nor changes during the study period of 6 months in the four groups, nor correlations between the results of quantitative sensory testing and the temperature sensation questionnaires. As stated in our methods, we only used a subset of the original 21 dimensions of the questionnaire, which was sensitive to sex-differences. Therefore, the questions we used might not be suitable for measuring sex-differences, or our cohort might have been too small" (pp. 18, 19, ll. 453-459).

Minor issue:

1. One of the highlights was “In persons assigned female at birth in the absence of hormone treatment, temperature detection thresholds were lower than in persons assigned male at birth”. However, in the first paragraph of the discussion, the authors indicated that this result was in accordance with previous literature (citing six studies), which makes it seem more like a confirmation of existing findings results rather than a novel highlight.

We completely agree with the reviewer and excluded this statement from the highlights section (p. 3, ll. 68-69).

2. In the discussion section and conclusion, the authors indicated that trans women’s treatment were “estrogen and testosterone blockers”. According to Table 2, trans women received Estradiol and Cyproteroneacetate (testosterone blocker) for gender-affirming hormone therapy. It would be more precise to use the singular form of “blocker” since there was only one kind of testosterone blockers.

Thank you for this useful comment! We totally agree and changed the wording in the whole manuscript to accurate descriptive language (see also Reviewer #1’s 2nd comment) and speak of “testosterone” and “estradiol and cyproterone acetate” (e.g., p. 2, ll. 58-63) when describing the treatment of our patients.

3. In the Figure 2a, the authors should denote what the triangle and the square represent.

Thank you also for this note! We changed the triangles in Figure 2a into squares since the x-axis already clearly defines the two groups.

Reviewer #3:

This study investigates sex hormone influences on thermosensation in transgender individuals undergoing GAHT. The study design is longitudinal with three data points (0, 3 and 6 months) with an appropriate cisgender control group. Thermosensation was assessed using a quantitative sensory testing protocol that included the assessment of detection and pain thresholds, as well as a questionnaire. The main finding is that feminizing GAHT in trans women leads to reduced detection thresholds, comparable with those observed in cisgender women compared to cisgender men.

We thank the reviewer very much for reviewing our manuscript!

There is no sample size justification or prior power analysis reported. Effect sizes are not reported. There are several null effects, but it’s not possible to have an insight on whether these are true null effects or the study is underpowered. Equivalence testing or Bayesian statistics would provide some insight on how to interpret the null effects. Correction for multiple comparisons is not mentioned.

We thank the reviewer very much for this important input and the excellent remarks and suggestions regarding the statistics of our study! An *a priori* power analysis was not conducted, this had several reasons: First, in addition to temperature perception, in our cohort of transgender persons (Hormones and Health Study, Franz et al. 2025 BMJ open) also other, mainly metabolic and cardiovascular, effects of hormone therapy are being studied, and we were not able to calculate estimated effects for all assessed parameters, as for several there were no good data to assess estimated effects. Second, the number of eligible participants could not be reliably estimated in advance, particularly as the already relatively rare condition of gender incongruence is frequently accompanied by psychiatric comorbidities which may influence the willingness to participate in a clinical study. Thus, we planned the study with intention to include as many patients as we could find to participate and then run the analyses. We are now transparently explaining this approach to the reader in the Methods section:

“An a priori power analysis was not conducted, as the expected effect sizes could not be estimated and the number of patients willing to participate could not be determined in advance. Instead, all patients who consented to participate during the study period and met the inclusion criteria were enrolled.” (p. 10, ll. 240-243).

Furthermore, we have now added a paragraph to the Methods part when and how we corrected for multiple comparisons:

“Global tests for the two a priori defined primary outcomes - temperature detection thresholds cold and warm - were conducted without multiple-comparison correction. For post-hoc pairwise comparisons, p-values were adjusted using the Bonferroni correction by dividing the nominal alpha level (0.05) by the number of pairwise tests (3), resulting in an adjusted significance threshold of $p < 0.0167$.” (p. 10, ll. 254-258).

Also, we now included the effect sizes (Cohen’s d, Kendall’s W) for all of our important results (temperature detection thresholds, pain sensation and temperature sensation questionnaire) in our manuscript.

- Specific instructions given to participants should be included (e.g., were the instructions the same as in the standardised QST manual or were they different). On this note, it seems that the protocol followed the German standardised QST protocol, but there is no explicit mention of whether it was the case or what are the specific differences that were introduced. For instance, were there familiarisation trials? The analyses says that 4 trials were analysed, but if the standardised version of the protocol was used, the first one is about familiarisation/demonstration, while the 2nd to 4th trials are the test trials to analyse

Thank you very much for this note. In many parts, our protocol followed the German standardized QST protocol, but not in all parts. We added a thorough description to the methods area with the specific instructions given to participants so that the reader can clearly see where instructions were the same as in the German standardized QST protocol and where not:

“All study staff underwent training and followed a standardized study protocol for both delivering verbal instructions to volunteers as well as performing the

quantitative sensory testing procedures. Procedures always took place at the same test location and during the same daytime and followed the same order. The thermode was placed at the palm as suggested by previous literature of the non-dominant hand and fixed with velcro tape. Participants were instructed to press a button with their dominant hand, as soon as they detected a change in temperature at the site of thermal stimulation. In the first set of measurements, the thermode's temperature decreased according to the method of limits with a rate of 0.3 °C/sec from its baseline temperature set at 32 °C until the participant pressed the button to measure cold detection thresholds. In the second set of measurements, the thermode's temperature increased accordingly to measure warm detection thresholds. Four measurements for cold perception and four for warm perception were conducted and the respective means were calculated and used for further analyses. Before the next sets of measurements, participants were instructed to press the button as soon as they felt a heat or cold induced pain. Again, starting from the skin indifference temperature of 32 °C, the thermode's temperature decreased/increased once at a rate of 1.5 °C/sec until the button was pressed or the lowest possible temperature of 0 °C/highest possible temperature of 50 °C was reached. We did not conduct further measurements of mechanical (pain) thresholds or measurements of different test sites.” (p. 9, ll. 212-230).

Minor comments

- *“In trans women, over the period of 6 months of GAHT, the thresholds for both “cold” ($p=0.004$) and “warm” ($p=0.045$) detection decreased (both $p>0.05$ in post-hoc tests).” This statement needs clarification: does this mean that the overall change over the 6 months period was statistically significant, but the specific time points (0 vs 3 and 3 vs 6) were not?*

Yes, this is correct. We clarified this for the reader by reporting the results of our analysis in clearer wording:

“In trans women, over the time period of 6 months of GAHT, the thresholds for both cold ($p=0.004$, Kendall's $W=0.63$) and warm ($p=0.045$, Kendall's $W=0.31$) detection decreased significantly. Measurements for the specific time points (0 vs. 3, 0 vs. 6, and 3 vs. 6 months) were not significantly different in the post hoc analysis (both $p>0.05$).” (p. 12, ll. 300-303).

- *There are instances of phrases like “temperature detection improved...” that should be avoided because they imply causality, which cannot be concluded from an observational design (i.e., no randomisation or placebo group). This could be rephrased to “was associated with...”*

Thank you for this good suggestion! According to the 2nd comment of Reviewer #1, we changed our wording throughout the manuscript and we are now speaking of “lowered” or “decreased temperature detection thresholds” and “increased sensitivity to changes in temperature”, rather than an “improvement” of perception (e.g., p. 2, ll. 58-63).

- *“discriminating “cold” and “warm” (page 9): the authors should be more careful with the use of the word “discrimination”, as that is a specific term associated with 2 AFC/IFC tasks, where the participants need to discriminate between two stimuli. Instead, the task used by the author is about “detection” of a single stimulus.*

Thank you for this comment! We changed the wording throughout the whole manuscript (e.g., p. 12, ll. 295-299).

- *In figures 2, it’s not obvious what the first and second rows of results represent. It took me quite a while to figure out that they are WDT (first row) and CDT (second row) based on the color of the dots*

Thank you for this notion! In order to make the Figure 2 clearer, we added the labels “warm detection threshold” and “cold detection threshold” to the y-axis and hope that it makes the Figure easier to understand. We changed similar Figures in the Supplement accordingly.

- *It would be more complete to also see a figure about cold and heat pain thresholds. Also the non-significant results about pain thresholds should be reported as they could be useful for future meta-analyses*

We thank the reviewer for this excellent suggestion! We added a respective Figure to the Supplement (Suppl Figure S4) and added on top a Figure of the results of the temperature sensation questionnaire (Supl. Figure S5).